# Whole Genome Sequencing of Avian Pathogenic *Escherichia coli* Causing Bacterial Chondronecrosis and Osteomyelitis in Australian Poultry

**DOI:** 10.3390/microorganisms11061513

**Published:** 2023-06-06

**Authors:** Max L. Cummins, Dmitriy Li, Aeman Ahmad, Rhys Bushell, Amir H. Noormohammadi, Dinidu S. Wijesurendra, Andrew Stent, Marc S. Marenda, Steven P. Djordjevic

**Affiliations:** 1Australian Institute for Microbiology and Infection, University of Technology Sydney, Ultimo, NSW 2007, Australia; max.cummins@uts.edu.au (M.L.C.);; 2The Australian Centre for Genomic Epidemiological Microbiology, University of Technology Sydney, Ultimo, NSW 2007, Australia; 3Faculty of Science, The University of Melbourne, Parkville, VIC 3010, Australia; 4Gribbles Veterinary Pathology, Clayton, VIC 3168, Australia

**Keywords:** avian pathogenic *E. coli* (APEC), ST95, ST117, ST57, ST69, bacterial chondronecrosis, bacterial osteomyelitis, microbial genomics, lameness, poultry production

## Abstract

Bacterial chondronecrosis with osteomyelitis (BCO) impacts animal welfare and productivity in the poultry industry worldwide, yet it has an understudied pathogenesis. While Avian Pathogenic *Escherichia coli* (APEC) are known to be one of the main causes, there is a lack of whole genome sequence data, with only a few BCO-associated APEC (APEC_BCO_) genomes available in public databases. In this study, we conducted an analysis of 205 APEC_BCO_ genome sequences to generate new baseline phylogenomic knowledge regarding the diversity of *E. coli* sequence types and the presence of virulence associated genes (VAGs). Our findings revealed the following: (i) APEC_BCO_ are phylogenetically and genotypically similar to APEC that cause colibacillosis (APEC_colibac_), with globally disseminated APEC sequence types ST117, ST57, ST69, and ST95 being predominate; (ii) APEC_BCO_ are frequent carriers of ColV-like plasmids that carry a similar set of VAGs as those found in APEC_colibac_. Additionally, we performed genomic comparisons, including a genome-wide association study, with a complementary collection of geotemporally-matched genomes of APEC from multiple cases of colibacillosis (APEC_colibac_). Our genome-wide association study found no evidence of novel virulence loci unique to APEC_BCO_. Overall, our data indicate that APEC_BCO_ and APEC_colibac_ are not distinct subpopulations of APEC. Our publication of these genomes substantially increases the available collection of APEC_BCO_ genomes and provides insights for the management and treatment strategies of lameness in poultry.

## 1. Introduction

Avian pathogenic *E. coli* (APEC) cause colibacillosis, a disease inflicting significant economic losses in commercial poultry production by reducing bird growth rates, layer productivity, and increasing mortality and meat condemnation [1]. The disease manifestations are diverse, but include colisepticemia, cellulitis, peritonitis, salpingitis, and bacterial chondronecrosis and osteomyelitis [2,3,4]. Although APEC are primary pathogens, host factors can play an important role in the susceptibility of birds to infections caused by these pathogens [1,5].

Bacterial chondronecrosis with osteomyelitis (BCO) is an important cause of lameness and a major animal welfare issue in the poultry industry [6]. This condition is predisposed by the formation of osteochondrotic clefts among the chondrocytes of susceptible growth plates as birds gain weight during their short production life cycle. Opportunistic bacteria that spread systemically can colonize these lesions and cause necrotic infection foci. *E. coli* commonly causes BCO, but several viral and bacterial pathogens are also linked with the condition [6,7,8]. Although the pathological events that precede *E. coli*-associated BCO remain poorly understood, gut insults caused by viral pathogens such as avian reovirus and chicken astrovirus has been implicated as a means of bacterial translocation into the circulatory system [1]. Similarly, APEC may cause lesions in the lungs following inhalation of airborne fecal particles, enabling their access to the blood. Regardless of the initial route, subsequent adherence of blood-borne *E. coli* to the vascular endothelium is a key event that enables subsequent invasion and infection of osteochondral tissues. The proximal growth plate of the tibiotarsus and femur are the most common sites of infection due to the nature of the vasculature that nourishes these sites [9].

It is unknown if *E. coli* that cause invasive diseases such as colibacillosis and BCO share genetic similarities. Genomic data regarding the *E. coli* strains that have been implicated in causing BCO are almost non-existent, with only three genomes available in public databases (https://pubmed.ncbi.nlm.nih.gov/34077848, accessed on 1 February 2023). The carriage of ColV F virulence plasmids is a defining feature of APEC [1,10,11] and is considered important in the causation of colibacillosis, although the role of ColV F plasmids may be clouded by the observation that faecal *E. coli* from healthy poultry are known to carry ColV plasmids [12]. However, like most extraintestinal pathogenic *E. coli* (ExPEC) in humans, APEC have a fecal origin. Therefore, it is not surprising that ColV plasmids are found in *E. coli* that colonize healthy poultry. In an avian experimental challenge model of APEC disease, a non-pathogenic commensal *E. coli* of avian origin gained the ability to colonize the murine kidney, reproduce in human urine, and cause death in chicken embryos upon acquisition of a ColV plasmid [13]. Given the multitude of virulence-associated genes (VAGs) that colocalize on most ColV F virulence plasmids, including several iron operons linked to iron acquisition, it is likely that APEC that cause BCO carry ColV virulence plasmids.

*E. coli* genomic sequences have been determined for organisms that have been recovered from cases of colibacillosis in Australia [14]. Salpingitis has been characterized to a lesser extent [15], but APEC associated with other invasive bacterial disease such as BCO remain poorly characterized. In this study, we generated genome sequences of 205 BCO-associated *E. coli* (APEC_BCO_), a collection that, to our knowledge, is the largest of its kind to date. We also provide a comprehensive pangenomic and phylogenomic analysis of these genomes. Additionally, we compared the BCO genomes with a collection of contemporaneous and geographically proximal genomes of *E. coli* causing colibacillosis (APEC_colibac_).

## 2. Materials and Methods

### 2.1. Sample Collection and Bacteriology

A collection of 214 APEC_BCO_ strains were sourced from deceased or culled broiler chickens which exhibited lameness, a symptom typical of BCO. The samples were collected between 2015 and 2016 from birds ranging in age from 7 to 35 days across 20 commercial broiler flocks (18/20 flocks were breed ‘Ross’, 2/20 flocks were breed ‘Cobb’) and thirteen farms in Victoria, Australia. A thorough postmortem was performed, and swabs were collected for bacteriology to isolate strains and confirm their species identity, as described in ref. [7].

As multiple collections and subcollections are described in this manuscript, please note the following: subsequent to DNA extraction, whole genome sequencing, and quality control (described below), 205 genomes remained after 9 genomes were removed due to low-quality sequence data. Please also note that of these 205 APEC_BCO_ genomes, a representative cohort of 66 genomes (APEC_REP-BCO_) was generated, which only included isolates from a single bone site per bird. This was required for statistical analysis, as the full APEC_BCO_ collection includes multiple isolates from different anatomical sites within the same bird. See *Pangenomics* section of the methods for more details on the APEC_REP-BCO_.

### 2.2. DNA Extraction, Whole Genome Sequencing and Assembly

Genomic DNA was extracted using the ISOLATE II Genomic DNA Kit (Bioline, Alexandria, NSW, Australia) following the manufacturer’s instructions and stored at −20 °C. Template DNA (0.5 ng) was used to prepare libraries using Nextera^®^ DNA library preparation kits, as previously described [16]. Whole genome sequencing of strains was performed using an Illumina HiSeq^®^ X Ten sequencer (San Diego, CA, USA) to generate 150 bp paired end reads. The genomic read sets were made publicly available under the Bioproject number PRJNA970052. The accession numbers for the read sets associated with individual strains are available in Appendix A.

### 2.3. Comparative APEC_colibac_ Collection

The genomes of 95 APEC from poultry with colibacillosis (APEC_colibac_) [14] were compared to the genomes of the APEC_BCO_ strains generated here. Sequence reads for these genomes (Bioproject number PRJNA479542) were accessed and processed alongside the APEC_BCO_ genomes.

### 2.4. Genomic Assembly and Assembly Stats

The sequence reads were filtered using fastp [17] v0.20.1 using default settings. All the subsequent steps involving sequence reads utilized these filtered reads. The genome assembly was performed using Shovill (https://github.com/tseemann/shovill) v1.0.4 using the parameter “--minlen 200”. All the analysis pipelines are provided as open-source scripts. Documentation of our in silico workflows and data visualizations are available at https://www.github/maxlcummins/bonechook. Many of the analyses below were performed using a genomic analysis pipeline available at https://www.github.com/maxlcummins/pipelord.

### 2.5. Quality Control

Genomic assembly statistics were generated using assembly-stats (https://github.com/sanger-pathogens/assembly-stats) v1.0.1. The assemblies were required to be of a length within 5% of that observed for *E. coli* within the RefSeq database: between 4,182,000 and 6,274,000 bp. Genome contiguity was assessed based on the assembly statistics, with assemblies required to have fewer than 800 scaffolds and an N50 of ≥15,000.

Genome completeness was assessed using CheckM v1.2.0 [18], a tool which counts and characterizes a suite of phylogenetic marker genes to calculate genomic completeness and contamination scores. The strains were required to be at least 90% complete and exhibit less than 10% contamination. The assemblies were also required to carry all seven genes required based on the Achtman MLST scheme, and MLST data were generated using mlst (https://github.com/tseemann/mlst) v2.19.0 using default settings. A total of 205 APEC_BCO_ genomes passed our quality control criteria, as well as all 95 APEC_colibac_. The scripts used to filter the strains based on the described quality control metrics described above are available at https://www.github/maxlcummins/bonechook.

### 2.6. Strain Typing

A comprehensive genomic analysis was performed to determine: (i) the *fimH* (fimbrial adhesin) type, (ii) the e-serotype (lipopolysaccharide and flagella types), (iii) clermont phylogroups, (iv) multilocus sequence type (Achtman scheme), and (v) plasmid multilocus sequence type (pMLST). The tools utilized included: (i) fimtyper [19] v1.1, (ii) ECtyper [20] v1.0.0, (iii) Clermontyper [21] v20.3 and EZClermont [22] v0.0.7, (iv) mlst (https://github.com/tseemann/mlst) v2.19.0, and (v) pMLST [23], respectively. All tools were run using default settings. The command lines used to run each tool are available via the aforementioned pipeline (https://github.com/maxlcummins/pipelord). Novel (Achtman scheme) multilocus sequence types were determined following the sequence being uploaded at Enterobase [24], from which core-genome MLST (cgMLST) and associated heirarchical clutering data were also obtained.

### 2.7. Genotyping

We screened for genes of interest including virulence-associated genes (VAGs), IS elements, plasmid replicons using VFDB [17], ISfinder [25], and Plasmidfinder [23] databases in combination with abricate (https://github.com/tseemann/abricate) v1.0.8. The exact versions of these databases, as well as a custom database featuring additional genes of interest that were not included in those aforementioned databases, are available at https://www.github/maxlcummins/bonechook. We also screened for AMR-associated genes and SNPs using Abritamr [26] v1.0.11, a wrapper for the AMRfinderplus pipeline v3.10.24 [27], using database version 2022-10-11.2.

### 2.8. Pangenomic Analysis

For the purpose of genome-wide association studies, we conducted pangenomic analyses on the collection of APEC_BCO_ and APEC_colibac_ strains. It is important to note that, for this analysis and the other statistical analyses described below, a representative subset of APEC_BCO_ genomes (referred to as APEC_REP-BCO_) were utilized. In this subset, only one strain per bird was included. Appendix A details the strains that were included in this category. To determine which strains to include in this cohort, the strains collected from non-bone sites were excluded, and a single strain from each bird was selected randomly using the R function ‘slice_sample’ from the package ‘dplyr’ (version 1.0.10). The quality control markdown file available at the GitHub repository associated with this manuscript provides a programmatic workflow that details this process. The pangenomic analysis was performed using Prokka [28] v 1.14.6 (default settings) and Panaroo [29] v1.2.10. The Snakemake [30] pipeline detailing this analysis is available at https://github.com/maxlcummins/pipelord/blob/main/workflow/Treebuild_panaroo.smk.

Subsequent genome-wide association studies were performed using Scoary [31] v1.6.16 (parameters: ‘--no_pairwise --threads 10 -s 4 --correction BH - --*p*_value_cutoff 1E-5’). Further genomic contexts of hypothetical proteins of interest were explored using MEGABLAST [32]. For this, hypothetical protein sequences were extracted from the Panaroo-derived pangenome reference assembly and searched against the NCBI nucleotide database using default settings (Access date 25 January 2023). The search results were investigated manually and the first *E. coli* sequence exhibiting a maximal E value was selected as a target for context analysis. An annotated assembly file (suffix ‘.ape’) was then downloaded from the associated nucleotide entry, and an alignment was performed using SnapGene software (https://www.snapgene.com) v4.1.9 before determining the location of the hypothetical protein of interest relative to proximal annotations. An additional scoary analysis was also performed using the same pangenome and parameters but included the argument ‘-r APEC-REP-BCO.csv’, where the indicated file contained a list of representative APEC_BCO_ strains, including only a single strain from each bird.

### 2.9. Non-Metric Multidimensional Scaling

We used the R packages vegan v2.6 [33], MASS v7.3 [34], ellipse v0.4.3, and grid v4.2.1 to generate and visualize non-metric multidimensional scaling (NDMS) of virulence and resistance traits among the genomes under investigation. We reduced the dimensionality among these profiles to both a two and three-dimensional space, and since the stress scores resulting from both analyses were comparable, we presented the two-dimensional reduction for ease of viewing. NDMS visualizations are available in Appendix A, and the associated stress scores for these NMDS analyses were 14.82% and 15.38%.

### 2.10. ColV Plasmid Typing

We modified a criteria defined by Liu et al. [35] to identify strains which carried ColV plasmids. This involved screening for the following ColV marker genes: (i) *cvaABC* and *cvi* (the ColV operon), (ii) *iroBCDEN* (salmochelin operon), (iii) *iucABCD* and *iutA* (aerobactin operon), (iv) *etsABC*, (v) *ompT* and *hlyF*, and (vi) *sitABCD*. Strains carrying VAGs from three or more of these groups, as well as the an IncF replicon, were classified as carrying a ColV plasmid. The tool utilized for ColV typing is available at https://github.com/maxlcummins/abricateR. Statistical analyses of ColV carriage rates among both the APEC_BCO_ and APEC_colibac_ collections were performed using the Chi-square test (α = 0.05).

### 2.11. Phylogenetic Analysis

Multiple approaches were undertaken to phylogenetically characterize the genomes under investigation. Phylogenetic trees were generated using a pangenome-derived core detailed at https://github.com/maxlcummins/pipelord/blob/main/workflow/Treebuild_panaroo.smk. Briefly, the characterization involved genome annotation with Prokka [28] v 1.14.6 (default settings), derivation of a core-genome via pangenomic analysis with Panaroo [29] v1.2.10, extraction of variable sites using SNP-sites v2.4.1 [36] and tree building using IQtree [37]. For the determination of pairwise SNP distances, we utilized SKA v1.0 [38], a method useful for high-resolution phylogenetic analyses. We utilized the commands ‘ska fastp’ and ‘ska distance’. The former command used default settings, and the latter used the parameters ‘-s 100 -i 90’ to generate pairwise SNP distances and identify clusters of genomes which differed by 100 or fewer SNPs and which shared 90% of split kmers. The Snakemake pipeline detailing this workflow can be found at https://github.com/maxlcummins/pipelord/blob/main/workflow/rules/phylogenetics/splitkmeranalysis.smk.

Lastly, as outlined earlier, cgMLST and associated HCC data from Enterobase [24] was used to cross-validate SNP distance metrics between APEC genomes. This groups genomes into hierarchical clusters (HCs) which allow hierarchical investigation of phylogenetic granularity based on core genome similarity (HC0 up to HC2350). Genomes which share an HC2 level differ by ≤2 genes in the cgMLST scheme, for example. More information is available at https://enterobase.readthedocs.io/en/latest/features/clustering.html (accessed 1 January 2023).

## 3. Results

### 3.1. Phylogenetic Analysis

Diverse STs were identified among the 205 APEC_BCO_ genomes, 29 of which were known, and four of which were novel (Table 1). The novel STs were designated ST11026 (strain FH3) (Clonal Complex 10), ST13151 (strain FH173), ST13152 (strains FH166 and FH167) (Clonal Complex 117), and ST13296 (FH118) in Enterobase (Clonal Complex 69). All eight phylogroups were represented in the BCO collection. The most predominant were phylogroups G, B2, E, and D. The top four sequence types (STs) identified in the BCO collection were ST117, ST95, ST69, and ST57. Together, these four STs represented almost 72% of the APEC_BCO_ collection (Figure 1) and ExPEC, with ST95, ST69, and ST117 encompassing lineages that cause extraintestinal disease in humans. Within these sequence types, especially those that were most prevalent, we identified a total of 22 *fimbrial* adhesin (*fimH*) types, indicating a diversity of sublineages within the sequence types (Table 1).

### 3.2. F Plasmid Carriage

F plasmids, especially ColV-like plasmids, are a characteristic feature of APEC associated with colibacillosis. In our study, we found that almost all isolates (202/205; 98.5%) in the BCO collection carried an F replicon. Using the criteria defined by Liu et al., (2018) [35], we determined that most (87.3% [179/205]) F plasmids in the BCO *E. coli* isolates were ColV-like plasmids. Plasmid typing revealed the presence of 20 IncF replicon sequence types (RSTs) (Figure 2). The most common F RSTs were F24:A-:B1 and F18:A-:B1, which were identified in seven and nine STs, respectively, accounting for 58% (119/202) of IncF plasmids in the BCO collection.

### 3.3. Virulence Gene Carriage

Next, we sought to determine the carriage of virulence genes within the collections under analysis. HPI was identified in 56% (136/205) of genomes under study, particularly in phylogroup B2 strains with STs 95 and 135 and phylogroup D strains with ST69. Further analysis identified two major clusters of virulence genes within our BCO genomes (Figure 3). The first of these, Cluster 1, was a suite of virulence genes common across most APEC isolates, which included fimbrial adhesin *fimH* as well as ColV-associated virulence genes such as *cvaABC/cvi*, *iroN*, *iucD/iutA*, *traT*, *ompT*, *sitA*, *hylF*, and *iss*. All of these genes were observed in 78% (160/205) of the APEC_BCO_ collection, except *cvaABC*/*cvi*, which was detected in 135/205 (66%) of the strains. Notably, Cluster 1 also carried chromosomally associated *fyuA/irp2* genes, which are reliabe gene markers for the Yersinia high pathogenicity island (HPI).

Cluster 2 contained genes less frequently identified within the collection (≤66/205 strains [32%]), including adhesion/invasion loci *hek*, iron acquisition operon *ets*, and the ColBM-associated colicin genes *cmi/cbi/cma/cba*. Pyelonephritis-associated pilus (*pap*) genes were common in ST95, ST355 (phylogroup B2), and some ST69 (phylogroup D), ST57 (phylogroup E), and ST117 (phylogroup G) genomes. This cluster also included VAGs associated with NMEC-associated *E. coliI*, such as *neuC, ibeA* and the capsular (K) antigen genes *kpsM, kpsMT(II)*, and *kpsMT(III)*. These genes were predominantly detected in phylogroup B2 (ST95 and ST355) strains, as well as iron-acquisition operon *eit* and uropathogen-specific protein *usp*. This cluster also contained genes including *pic*, *vat*, *astA*, *iha* and *shiD*, which were variably present in APEC STs.

We also performed an additional GWAS on a representative subset of the APEC_BCO_ collection (referred to as APEC_REP-BCO_) to explore potential differences in the carriage of VAGs by STs recovered from multiple sites and STs recovered from a single site. We identified a total of eight genes, four of which were hypothetical proteins of unknown functions. The remaining four genes (specificity of each was 88.9%, sensitivity of each was 84.2%, OR of each was 42.7, and the Benjamini and Hochberg-adjusted *p* of each was <0.05, Appendix A) included *lsrB_2/lsrD_2* (annotated as autoinducer 2-binding protein and autoinducer 2 import system permease, respectively) and *rbsA_2/C_3* (annotated as ribose import ATP-binding protein and ribose import permease protein, respectively).

### 3.4. Infection Dynamics

To examine the APEC_BCO_ strains in more detail, we focused on infections spanning multiple sites and explored their strain sequence type. We found that isolates belonging to ST69, ST95, ST57, and ST117 infected multiple sites per bird (mean site count 3.57, 3.45, 3.29, and 2.67, respectively). No statistically significant difference was determined between these STs in this regard (ANOVA, α = 0.506). Additionally, STs ST1640, ST355, and ST752 infected a similar number of sites per bird (µ = 3.0 each); however, these STs infected far fewer birds in total (each one to two birds) than the other STs described (each infected seven birds or more). Overall, ST117 infected a greater number of birds than any other ST (n = 23); more than twice that of the next-closest ST, ST95 (n = 11), and more than three times that of the next most infectious STs: ST57 and ST69 (n = 7).

Of the 70 birds from which *E. coli* were isolated, 1/70 (1.4%) yielded three STs, 13/70 (18.6%) yielded two STs, and the remainder (56/70; 80%) yielded only a single ST. Of the 14 birds that were infected by multiple sequence types, two birds yielding two STs were infected by strains of the same clonal complex (i.e., differed by only one MLST allele), while the remainder were infected by multiple STs which did not share a clonal complex; evidence of polyclonal infections. A total of 250 pairs of genomes shared an ST and were sourced from the same bird. These were found to be highly clonal, with 70% (175/250) exhibiting zero SNPs, 18.4% (46/250) exhibiting one SNP, 7.2% (18/250) exhibiting two SNPs, and 2.8% (7/250) exhibiting three SNPs. One other strain differed by six SNPs (1/250 [0.4%]), a further two strains differed by nine SNPs (2/250 [0.8%]), and one strain differed by 45 SNPs (1/250 [0.4%]), the latter of which was also found to differ by between 10 and 20 cgMLST alleles.

### 3.5. Phylogenetic Comparisons with APEC_colibac_

A representative subset of the APEC_REP-BCO_ was selected and compared with a previous APEC colibacillosis (APEC_colibac_) collection [14] through a genome-wide association study. Similar to APEC_BCO_, a previously described collection of APEC_colibac_ derived from farms in similar geographic locations and within a similar time period was shown to consist of 30 STs, the most common of which were also ST117, ST95, and ST57. The APEC_BCO_ collection exhibited much higher representation of ST95 (38/205; 18%) than the APEC_colibac_ (8/95; 8%), and this was also reflected in the APEC_REP-BCO_ subset (18/66, 27%) which only included one isolate per bird. ST350, ST429, and ST973 were featured prominently in the APEC_colibac_ collections, but were notably absent (except for one ST350 isolate) from the APEC_BCO_ collection. However, these collections generally exhibited extensive overlap at the phylogroup and ST level, with phylogroups G, B2, and E represented across both collections (Appendix A). The phylogenetic analysis revealed that while some APEC_BCO_ and APEC_colibac_ genomes localized to separate branches in the broader phylogeny, there was no evidence to indicate that the APEC_BCO_ and APEC_colibac_ genomes were derivates of distinct lineages. Instead, phylogenetic subclades revealed numerous instances where closely related strains across the two collections were in close proximity within the phylogeny. To investigate this further, we performed additional high-resolution genomic analyses to determine the degree of relatedness among the STs observed in both collections.

For this analysis we focused on ST117, ST57, and ST95, as these were common to both collections and were in relatively high abundance. We compared the SNP distances between pairs of genomes of these sequence types within collections and between collections (Appendix A). Additional analyses were performed on all pairwise SNP distances between members of the same ST across the APEC_BCO_ and APEC_colibac_ collections to maximize the number of pairwise distances and identify the most closely related combinations. For example, a total of 1364 pairs of ST117 existed between the collections (62 APEC_BCO_ ST117 strains × 22 APEC_colibac_ strains = 1364 pairs). SNP distances both within and between the collections varied greatly, from as low as 10 (ST95) to as high as 6233 (ST57). Among pairs of genomes spanning both collections, 57/207 ST57, 36/304 ST95, and 14/1364 ST117 pairs exhibited ≤100 SNPs (Figure 4). Of these, all ST117 and ST95 pairs differed by 20 or fewer cgMLST alleles, and all ST57 differed by 50 or fewer cgMLST alleles. Additionally, a further 9/207 ST57, 33/304 ST95, and 9/1364 ST117 genome pairs exhibited ≤50 SNPs, while 20/304 genome pairs of ST95 differed by ≤25 SNPs. These SNP distances were also supported by small cgMLST allelic distances. For example, all pairs separated by 50 or fewer SNPs also differed by ≤20 cgMLST alleles, while 18/20 ST95 pairs which differed by ≤25 SNPs exhibited an allelic distance of ≤10. Furthermore, all genomes which shared a ST and differed by 100 or fewer SNPs differed by ≤20 cgMLST alleles.

### 3.6. Accessory Genome Clustering and Genome-Wide Association Studies

We performed non-metric multidimensional scaling (NDMS) on the virulence profiles of these isolates to explore potential clusters. This approach did not reveal discretely clustered strains based on their origin of collection (APEC_BCO_ or APEC_colibac_) or sequence type, except for ST95 strains, which clustered neatly based on their reduced virulence profiles (Appendix A). Notably, in contrast to the virulence NDMS analysis, the same approach using clustering based on strain ARG profiles revealed strains clustered in two-dimensional space in accordance with the collection’s origin and the presence of a Class 1 integrase gene (Appendix A).

To further explore whether ARG or virulence traits were over or underrepresented within either APEC_BCO_ or APEC_colibac_, we performed a genome-wide association study. To avoid biasing statistical tests, the study was performed on the APEC_REP-BCO_ (see methods) rather than the full APEC_BCO_ collection. We found no evidence of genes that were overrepresented in APEC_REP-BCO_ genomes. A total of 10 genes were identified which were under-represented in APEC_REP-BCO_, all of which were either AMR genes or were co-carried at rates equal or close to AMR genes (Appendix A). For example, of the 10 genes identified as overrepresented in the APEC_colibac_ genomes, 6/10 were associated with integrons (*xerC*, also known as *intI1*) and AMR traits (*emrE*, *folP_2*, *tetA*, *tetR*, and *dhfrI*). The remaining four traits were hypothetical proteins of unknown function; however, subsequent BLAST analysis revealed their association with AMR loci. For example, group_3787 and group_6608 were integron associated; the former was located 127 bp downstream of *sul1* in an *E. coli* plasmid from the USA (Accession: CP097713.1), and the latter was identified as a transposase located 223 bp downstream of *intI1* Accession: LC257591) in an *E. coli* plasmid from a human (country unknown). In contrast, the hypothetical proteins group_6432 and group_4547 were found to be associated with tetracycline operons. Group_6432 was identified 32 bp from *tetA*(4)/*tetR* operon (Accession: MK962306) in an *E. coli* plasmid from China, and group_4547 was identified as 31 bp from a *tetA*(6)/*tetR* loci in another *E. coli* plasmid from China (Accession: ON934557).

### 3.7. Carriage of ColV-like Plasmids

Finally, we investigated the carriage rates of ColV plasmids within and between the APEC_BCO_ and APEC_colibac_ collections. Our analysis determined that 269/300 (90.0%) carried such plasmids, of which 179/205 (87.3%) were APEC_BCO_ and 90/95 (94.7%) were APEC_colibac_. This difference was not found to be statistically significant (χ2 test *p* = 0.07839). Similarly, an analysis of ColV carriage among the representative APEC_REP-BCO_ (see methods) and APEC_colibac_ revealed that the former exhibited 90% ColV carriage (60/66). This difference was also was not statistically significant (χ2 test, *p* = 0.5292).

## 4. Discussion

Here we present the largest and most detailed genomic and phylogenomic analysis of BCO-associated APEC, shedding new light on this previously poorly characterized subset of avian pathogens. Additionally, we leveraged high-resolution phylogenetic data to provide insights into polyclonal APEC_BCO_ infections, including not only those caused by phylogenetically distinct strains of the same sequence type, but also those caused by multiple sequence types.

The prior absence of genomic data for APEC_BCO_ has precluded high-resolution analysis to determine its relatedness, or lack thereof, to APEC_colibac_. Overall, the traits of APEC_BCO_ reflect those of APEC causing colibacillosis. For example, common STs include 117, 95, 57, and 69—sequence types we (Cummins et al., 2019) and others [2,39] have documented as APEC-causing colibacillosis from Australia and overseas. The extensive overlap at the sequence type level indicates that these two populations of APEC share relatively close evolutionary histories. Some members of the same STs, both within and between collections, differ by their *fimH* types, demonstrating a diversity of sublineages within these STs capable of causing extraintestinal infections in poultry. Our subsequent high-resolution analyses revealed that these strains can differ by thousands of SNPs, supporting this contention. While it is difficult to quantify the divergence within and between these two collections attributable to sampling, we also found closely related pairs of samples from both collections. Intercollection comparisons of ST95, ST117, and ST57 genomes revealed 107 genome pairs (comprising 69 strains) that differed by 100 or fewer SNPs, while some differed by as few as 10 SNPs. Therefore, many of these APEC strains causing BCO are clonally related or closely related to those causing colibacillosis. We did not collect specific information on the rearing conditions, feed supplementation, or other management practices that may explain the high level of genomic relatedness observed amongst isolates. However, at the time of recruiting the farms for the APEC_colibac_ and the APEC_BCO_ studies, there were only a few large commercial poultry producers in Australia. Typically, these companies source their chicks from a small number of dedicated hatcheries and place the birds in conventional broiler farms. It is therefore likely that these observations of close relatedness between members of the two collections can be explained by the local uniformity of intensive farming in the industry.

ColV-like plasmids and their associated virulence genes and fitness factors have been identified globally as hallmarks of APEC [2,14,40]. While we previously reported extensive carriage of ColV in APEC_colibac_ [14], here we present strong evidence of the presence of extensive carriage of ColV-like plasmids among APEC causing BCO. Similar to the APEC_colibac_ collection, our APEC_BCO_ genomes carried a diversity of plasmid IncF RSTs, but some spanned multiple STs and were common in the most frequently isolated APEC STs. This suggests that these IncF plasmids also play important roles in APEC_BCO_ virulence. Additionally, IncF RST F18:A-:B1 (also sometimes reported as C4:A-:B1) was the most prevalent plasmid variant detected within both APEC_colibac_ and APEC_BCO_ collections, indicating these two groups carry similar ColV plasmids (Appendix A).

We also highlight the similarity between the genotypic virulence profiles of APEC_colibac_ and APEC_BCO_ and demonstrate their overlap using NMDS. Our genome-wide association study found that the only genes which were disproportionately present in either group of APEC were those associated with class 1 integrons and tetracycline resistance operons (which often colocalize with these integrons). This discrepancy is readily explained by the selection criterion (class 1 integrase carriage) used to select APEC_colibac_ for whole genome sequencing in Cummins et al., (2019)—a criterion that was not applied for the selection of isolates in our APEC_BCO_ collection. However, the failure of our GWAS to detect any genes overrepresented in APEC_BCO_ indicates that it is unlikely their involvement in cases of BCO depends on the acquisition of specific virulence loci that distinguish them as a subset of APEC. However, we cannot rule out differences in gene expression that are necessary for BCO pathogenesis. Exploration of the genomic factors within BCO that may influence their capacity to aggressively invade their hosts was also conducted via our GWAS on genomes of STs recovered from multiple sites compared with those isolated from a single site. *lsrB* and *lsrD* have been shown to play key roles in quorum sensing and antibiotic susceptibility in APEC [41], and may also be involved in stimulating biofilm formation and enhancing cell motility via their roles in Autoinducer-2 transport [42]. In contrast, *rbsA* and *rbsC* are involved in ribose metabolism [43], though the relevance of carrying these genes (and the specific loci we identified as overrepresented) is less clear. It is possible that their overrepresentation in hyperinvasive STs is coincidental due to their physical association with the *lsr* variants we observed, based on identical patterns of co-carriage of genes from both operons (Appendix A).

While differences in sampling methodology introduce bias into our study, it is unlikely that this factor played a major role in influencing the degree of similarity we reported between these two populations of APEC genomes. Additionally, using this collection of APEC_colibac_ enabled us to capture the most geographically and temporally proximal collection for comparison with APEC_BCO_ strains. Note that while we have provided some Appendix A on AMR gene carriage, it is because of this bias that we do not draw comparisons between the AMR and AMR-associated traits of these collections. However, it is notable that of the 256 APEC_colibac_ isolated in our previous study [14], 95/256 (37.1%) carried a class 1 integrase; in contrast, only 6/66 (9.1%) of our representative APEC_BCO_ isolates carried this gene. Class 1 integrons in poultry and more broadly are often associated with sulphonamide and trimethoprim resistance [14,44], both of which are antibiotics used to raise poultry in Australia. It is therefore possible that strains causing colibacillosis are more commonly resistant to these antibiotics than APEC_BCO_. One explanation may be reflected in the ability of antibiotic compounds to reach joint sites compared to other organ sites commonly affected in cases of colibacillosis, but this speculative. Alternatively, BCO lesions may have a predilection for forming in cartilage channels that are relatively poorly perfused when inadequate antibiotic distribution occurs. Additionally, information regarding historic use of antibiotics at the farms from which these isolates were collected is unavailable; a factor which would greatly influence the resistance traits observed. It is also possible that historic antibiotic usage on farms that contributed APEC_colibac_ was higher than those from which APEC_BCO_ were collected.

In conclusion, our genomic analyses were unable to pinpoint any notable differences when comparing APEC_BCO_ and APEC_colibac_. Furthermore, our studies provide evidence that, despite a limited sample size, APEC_BCO_ and APEC_colibac_ can be phylogenetically clonally related and exhibit extensive genetic similarity in the presence of virulence genes.

## Figures and Tables

**Figure 1 microorganisms-11-01513-f001:**
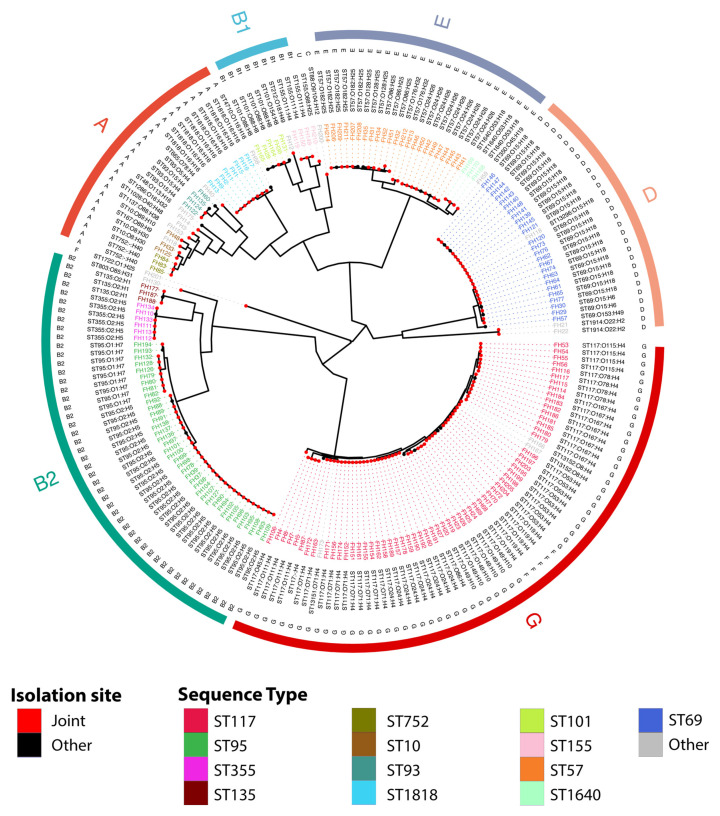
Pangenome-derived core-genome phylogeny visualizing relatedness of 205 APEC_BCO_ strains. Tip points are colored based on the site of isolation for a given strain, while primary tip labels adjoined with dotted lines indicate strain names and are colored based on sequence type. The next outermost band details the ST and e-serotype combinations. Phylogroups for individual strains are denoted on the outermost lettered band adjacent to broad, colored bars highlighting the general correspondence of clades and phylogroups. The tree is midpoint rooted.

**Figure 2 microorganisms-11-01513-f002:**
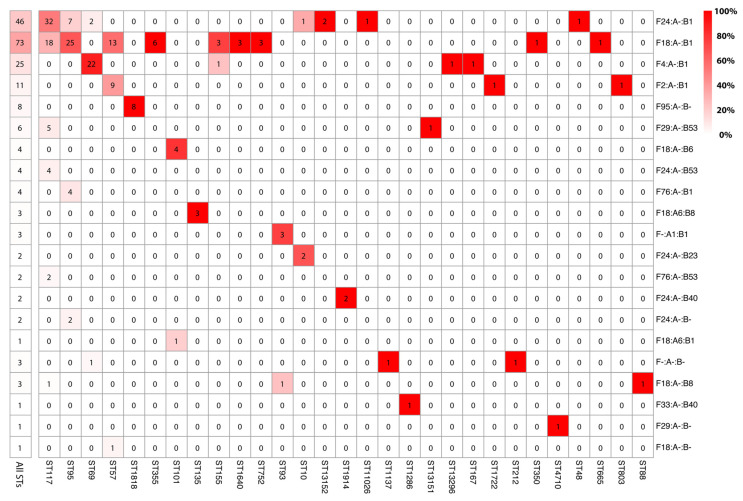
Carriage of IncF RSTs by MLST. Cell values represent the frequency (as a count) with which an IncF RST was identified within a given ST. Cells are colored based on their column-wise representation within a given ST (as a percentage), where the shades of red and their corresponding percentages shown in the key (top right).

**Figure 3 microorganisms-11-01513-f003:**
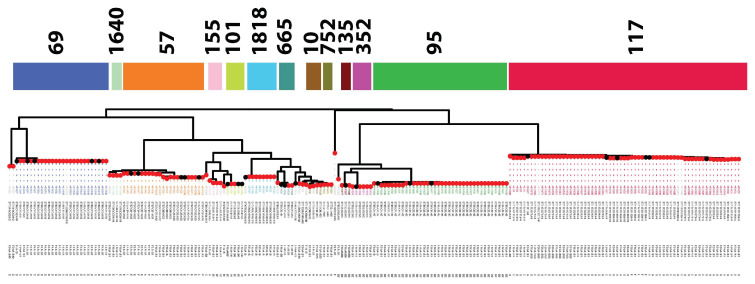
Virulence traits identified in APEC_BCO_ genomes. A pangenome-derived core-genome phylogeny is presented atop a table visualizing virulence genotypes. Tip points are colored based on the site of isolation for a given strain, while primary tip labels adjoined with dotted lines indicate strain names and are colored based on sequence type to allow viewers to follow the uppermost band denoting primary STs within the collection. The next outermost band details the ST and e-serotype combinations. Phylogroups for individual strains are denoted on the outermost lettered band, which details the best-viewed digital format. Below, virulence gene names are shown to the right of a given row alongside the frequency of a given gene (as both a count and percentage) across the APEC genomes. Virulence gene (row) order is clustered using strain-wise Euclidean distances of virulence profiles, such that genes which tend to co-occur are grouped together. The clusters were then highlighted and numbered sequentially (left).

**Figure 4 microorganisms-11-01513-f004:**
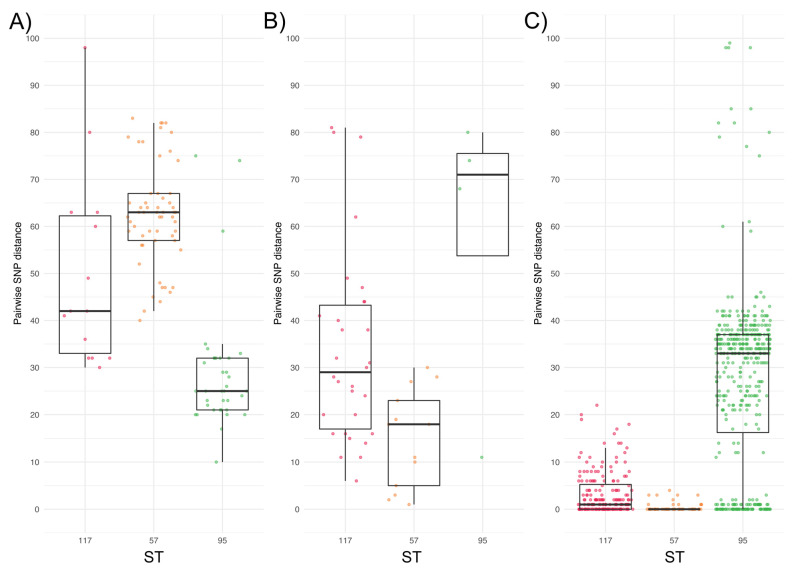
Box and jitter plot visualizing pairwise SNP distances (*y*-axis) between strains sharing a sequence type (*x*-axis). (**A**) Comparison between APEC_BCO_ and APEC_colibac_ collections, (**B**) within APEC_colibac_ collection, and (**C**) within APEC_BCO_ strains. Strain pairs of only three predominant STs are shown and colored by ST; ST117 (red), ST95 (yellow), and ST57 (green). Note that only strain pairs differing by 100 or fewer SNPs and those of ST117, ST57, and ST95 are included in this diagram—a similar figure containing all strain pairs within these three STs are shown in Appendix A.

**Table 1 microorganisms-11-01513-t001:** Frequency of predominant APEC BCO sequence types. This table presents the frequency of detected STs within APEC isolates, reported as counts and percentages in brackets. The phylogroup designations within each ST are based on ClermonTyper. * denotes a novel ST, ^^^ NT indicates that O antigen genes were not detected, and ^^^^ represents a novel *fimH* allele which most closely matches the denoted allele.

ST	Phylogroup	O-Types	H-Types	*fimH* Types	APEC_BCO_
117	G	NT ^^^, 111, 115, 119, 149, 167, 24, 45, 53, 71, 78, 86	10, 4	96, 917	62/205 (30.2%)
95	B2	1, 2	5, 7	27, 30, 526	38/205 (18.5%)
69	D	15, 153	18, 49, 6	27	25/205 (12.2%)
57	E	128, 176, 182, 24, 86	25, 26, 32	54, 158, 759	23/205 (11.2%)
1818	A	116	16	86	8/205 (3.9%)
355	B2	2	5	154	6/205 (2.9%)
101	B1	154, 88	8	86	5/205 (2.4%)
93	A	15, 5	4	32, None	4/205 (2.0%)
155	B1	111, 88	25, 4	121, 366	4/205 (2.0%)
1640	E	53	18	27	3/205 (1.5%)
10	A	8, 89	10, 30	54, None	3/205 (1.5%)
135	B2	2	1	8 ^^^^	3/205 (1.5%)
752	A	NT ^^^	40	24	3/205 (1.5%)
13152 *	G	8	4	97	2/205 (1.0%)
1914	D	22	2	550	2/205 (1.0%)
88	C	9/104	12	25	1/205 (0.5%)
11026 *	A	40	48	54	1/205 (0.5%)
1137	A	88	38	54	1/205 (0.5%)
1286	A	16	32	23	1/205 (0.5%)
13151 *	G	71	4	97	1/205 (0.5%)
13296 *	D	15	18	27	1/205 (0.5%)
11026 *	A	40	48	54	1/205 (0.5%)
167	A	89	9	None	1/205 (0.5%)
1722	F	1	25	153	1/205 (0.5%)
212	B1	18	49	38	1/205 (0.5%)
4710	A	116	16	86	1/205 (0.5%)
48	A	113	16	23	1/205 (0.5%)
803	B2	85	31	88	1/205 (0.5%)
665	A	78	4	30	1/205 (0.5%)

## Data Availability

The data presented in this study are available at https://www.ncbi.nlm.nih.gov/bioproject/PRJNA970052 (accessed on 2 June 2023). Additionally, the scripts involved in the generation, processing, and visualization of these data are available at https://github.com/maxlcummins/bonechook (accessed on 2 June 2023).

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
