# Peer review of "Whole Genome Sequencing of Avian Pathogenic Escherichia coli Causing Bacterial Chondronecrosis and Osteomyelitis in Australian Poultry"

_microorganisms, 2023, doi:10.3390/microorganisms11061513_

Round 1

Reviewer 1 Report

The authors did a great job analyzing 200+ sequences of E. coli isolated from bacterial chondronecrosis and osteomyelitis cases in Australia. There are some minor editing in the paper, and some important comments on the discussion. 

Lines 45-50. Expand on role of gut insult leading to bacterial translocation - Avian Reovirus, Chicken Astrovirus, etc. 

Line 58. Just for clarification "like all extraintestinal pathogenic E. coli (ExPEC) in humans," 

Line 82. Rewrite repeated words. "following. Following" .

Line 153. Modify to "collected from non-bone sites".

Line 210. Define ST

Lines 382-446. Results discussed in this paper are based in samples obtained between 2015-2016 from 20 commercial flocks in Victoria, Australia. No information is given if these commercial flocks are from the same farm(s), same company, nor from the same hatchery. Perhaps there is not enough variability (different companies, different farms, etc.) to extrapolate some of the findings to a larger universe. No comment is given on the treatment given to the farm litter on each cycle (i.e., reused, new litter each cycle, downtime), nor if the flocks were Raised Without Antibiotics, or conventional production, all of which might explain recycle of pathogens on the same environment. Comment on correlations of presence of VAG and plasmids with SPF virulence assessment and justify why this in-vivo method was not used. 

Quality of English Language was excellent. Only requiring some very minor editing. 

Author Response

Reviewer 1

The authors did a great job analyzing 200+ sequences of E. coli isolated from bacterial chondronecrosis and osteomyelitis cases in Australia. There are some minor editing in the paper, and some important comments on the discussion. 

We thank reviewer 1 for their efforts and for their compliments of our analysis

Lines 45-50. Expand on role of gut insult leading to bacterial translocation - Avian Reovirus, Chicken Astrovirus, etc. 

We changed the sentence on lines 48-52 accordingly (additions underlined):

Although the pathological events that precede E. coli-associated BCO remain poorly understood, adherence of blood-borne E. coli to vascular endothelium is a key event enabling subsequent invasion and infection of osteochondral tissues, with the proximal growth plate of the tibiotarsus and femur being the most common sites of infection due to the nature of the vasculature that nourishes these sites [10].

Although the pathological events that precede E. coli-associated BCO remain poorly understood, gut insult by viral pathogens such as Avian Reovirus and Chicken Astrovirus has been implicated as a means for bacterial translocation into the circulatory system [1]. Similarly, APEC may cause lesions on the lungs following inhalation airborne faecal particles, enabling their access to the blood. Irrespectively, subsequent adherence of blood-borne E. coli to vascular endothelium is a key event enabling subsequent invasion and infection of osteochondral tissues, with the proximal growth plate of the tibiotarsus and femur being the most common sites of infection due to the nature of the vasculature that nourishes these sites [10].

Line 58. Just for clarification "like all extraintestinal pathogenic E. coli (ExPEC) in humans," 

This change has been implemented on what is now line 64

Line 82. Rewrite repeated words. "following. Following" .

Second instance of “Following” replaced with “Subsequent to”, now on line 88.

Line 153. Modify to "collected from non-bone sites".

On line 166, “collected non-bone sites” changed to “collected from non-bone sites”

Line 210. Define ST

This is now defined on line 234

Lines 382-446. Results discussed in this paper are based in samples obtained between 2015-2016 from 20 commercial flocks in Victoria, Australia. No information is given if these commercial flocks are from the same farm(s), same company, nor from the same hatchery.

A total of thirteen farms contributed samples to this study – this information was mistakenly omitted from the methodology and has now been included (line 84). Thank you to Reviewer 1 for bringing this to our attention. Unfortunately, more detailed information about the relationships of these farms (i.e. their association with companies or hatcheries is unavailable).

Perhaps there is not enough variability (different companies, different farms, etc.) to extrapolate some of the findings to a larger universe. No comment is given on the treatment given to the farm litter on each cycle (i.e., reused, new litter each cycle, downtime), nor if the flocks were Raised Without Antibiotics, or conventional production, all of which might explain recycle of pathogens on the same environment.

Our metadata pertaining to farming practices is incomplete obfuscating the potential relevance of these factors to our observations. We have now acknowledged this as a limitation of the study on lines 446-454 and 505-508.

Comment on correlations of presence of VAG and plasmids with SPF virulence assessment and justify why this in-vivo method was not used. 

While we acknowledge the value of this analysis it is outside of the scope of this research article.

Reviewer 2 Report

Interesting manuscript. Congratulations.

Author Response

Interesting manuscript. Congratulations.

We thank Reviewer 2 for their efforts in reviewing the manuscript and for their comments

Reviewer 3 Report

Dear Authors,

you manuscript entitled "Whole genome sequencing of Avian Pathogenic Escherichia coli causing bacterial chondronecrosis and osteomyelitis in Australian poultry" was a detailed insight in the genome wide features of E. coli isolate collections from poultry diseases colibacillosis and joint infections. The identification of the phylogenetic lineages and STs involved and the definition of their genome wide similarities, plasmid and virulence carriage, led to a thorough characterization of these pathogenic E. coli groups that represent a very relevant contribution to the knowledge on these pathogens.

Author Response

Dear Authors,

you manuscript entitled "Whole genome sequencing of Avian Pathogenic Escherichia coli causing bacterial chondronecrosis and osteomyelitis in Australian poultry" was a detailed insight in the genome wide features of E. coli isolate collections from poultry diseases colibacillosis and joint infections. The identification of the phylogenetic lineages and STs involved and the definition of their genome wide similarities, plasmid and virulence carriage, led to a thorough characterization of these pathogenic E. coli groups that represent a very relevant contribution to the knowledge on these pathogens.

We thank Reviewer 3 for their efforts in reviewing the manuscript and for their comments

Reviewer 4 Report

The manuscript authored by Cummins et al reports their investigation on the isolation, whole genome sequencing and in-depth analysis of avian pathogenic E. coli that are associated with chondronecrosis and osteomyelitis (APECbco) in broiler chickens. The authors sequenced and examined the whole genomes of 250 APECbco isolates and generated high resolution phylogenetic analysis of APEC including sequence type diversity and carriage of virulence associated genes (VAGs). They have also performed genome wide association study with APEC which were isolated from multiple cases of colibacillosis (APECcolibac). The study is well planned and executed and the manuscript is very well written. The study provides very important information on the genetic characteristics of APECbco and their evolutionary relatedness to APEC that are associated with colibacillosis.

The following are minor questions:

1.     Section 2.1. Please include the type of broiler chicken breeds and age range from which samples were collected from.

2.     Was there a difference in the type of virulence genes carried between the APECbco sequence types which infected multiple sites as compared to those types that are associated with only a single site?

3.     Was there an association between breeds of broiler chickens and susceptibility to a particular sequence type of APECbco?

Author Response

Reviewer 4

The manuscript authored by Cummins et al reports their investigation on the isolation, whole genome sequencing and in-depth analysis of avian pathogenic E. coli that are associated with chondronecrosis and osteomyelitis (APECbco) in broiler chickens. The authors sequenced and examined the whole genomes of 250 APECbco isolates and generated high resolution phylogenetic analysis of APEC including sequence type diversity and carriage of virulence associated genes (VAGs). They have also performed genome wide association study with APEC which were isolated from multiple cases of colibacillosis (APECcolibac). The study is well planned and executed and the manuscript is very well written. The study provides very important information on the genetic characteristics of APECbco and their evolutionary relatedness to APEC that are associated with colibacillosis.

The following are minor questions:

  1. Section 2.1. Please include the type of broiler chicken breeds and age range from which samples were collected from.

This has now been included on line 83-85.

  1. Was there a difference in the type of virulence genes carried between the APECbco sequence types which infected multiple sites as compared to those types that are associated with only a single site?

We performed an additional GWAS to explore this and added our results on lines 286-294 and some discussion on this matter on lines 474-484. This analysis is a nice addition and we thank the reviewers for this suggestion. We also adjusted the methods on lines 183-186 to reflect this and an additional supplementary table with supporting data.

  1. Was there an association between breeds of broiler chickens and susceptibility to a particular sequence type of APECbco?

While this analysis would likely be worthwhile in future studies, we cannot undertake it in a statistically robust manner due to only a minority of samples being of a different breed.